# Fabrication of Superhydrophobic and Light-Absorbing Polyester Fabric Based on Caffeic Acid

**DOI:** 10.3390/polym14245536

**Published:** 2022-12-17

**Authors:** Xue Lei, Ailing Xie, Xinya Yuan, Xueni Hou, Jiaosheng Lu, Ping Liu, Zhonglin Xiang, Guoqiang Chen, Tieling Xing

**Affiliations:** 1College of Textile and Clothing Engineering, Jiangsu Engineering Research Center of Textile Dyeing and Printing for Energy Conservation, Discharge Reduction and Cleaner Production (ERC), Soochow University, Suzhou 215123, China; 2Jiangsu Lianfa Group, Nantong 226001, China

**Keywords:** caffeic acid, polyester fabric, superhydrophobicity, enol click chemistry

## Abstract

Caffeic acid (CA) was treated on the surface of polyester fabric (PET), and Fe^2+^ was used as an intermediate to form chelates with CA to increase the roughness of the polyester surface. With the addition of n-octadecyl mercaptan (SH), the mercapto group reacted with the carbon–carbon double bond of CA on the PET surface through enol click chemical reaction. Meanwhile, CA was polymerized under UV radiation, and thus CA-Fe-SH-PET was prepared. The introduction of SH with a long carbon chain reduced the surface energy of the PET, in order to endow the polyester fabric with a superhydrophobic/lipophilic function. Combined with XPS and FTIR tests, the new carbon–carbon double bond’s binding energy and vibration peak were found on the fabric surface, indicating that CA was adsorbed on the PET fabric’s surface. After adding SH, the double bond disappeared, demonstrating that SH and CA occurred a click chemical reaction and were grafted onto the PET fabric’s surface. The water contact angle (WCA) of CA-Fe-SH-PET was about 156 ± 0.6°, and the scrolling angle (SA) was about 3.298°. The results showed that the modified polyester had a robust superhydrophobic stability in washing, mechanical friction, sun aging, seawater immersion, organic reagent, and acid-base erosion derived from the good adhesion of polymerized CA (PCA). At the same time, the modified polyester fabric had good self-cleaning, antifouling, and oil–water separation performance. It was found that the CA-Fe-SH-PET fabric had unique photothermal conversion characteristics, which can convert the absorbed ultraviolet light into thermal energy, providing a local warming effect due to rapid heating and improving the transmission speed of heavy oil (engine oil and diesel). The CA-Fe-SH-PET fabric can further prevent the transmission of ultraviolet rays, and the UV resistance of CA-Fe-SH-PET fabric is far higher than the UV resistance standard. The preparation method is simple, fast, efficient, and environmentally friendly, and it has better a potential application value in the oil–water separation field.

## 1. Introduction

The lotus leaf effect and butterfly wings in nature are superhydrophobic effects [1,2]. Superhydrophobic properties are widely used in our daily life, such as in self-cleaning, antifouling, oil–water separation, anti-ice, anti-fog, anti-frost [3], and antibacterial applications. Superhydrophobic surfaces are generally achieved by increasing surface roughness and reducing surface energy [4]. In order to obtain the superhydrophobic function, some researchers have used silica particles to increase the surface roughness and long chain silane, resin, or fluorinated organic matter to reduce the surface energy of the material through spraying and other methods [5,6,7,8,9]. Researchers also coated diatomite on glass to achieve a micro/nanoscale surface and then further chemically modified the surface with fluorosilane to obtain a superhydrophobic surface [10]. However, superhydrophobic surfaces have poor durability and a short service life. When exposed to harsh environments, the surface is prone to being damaged. It has been difficult to apply in practice, and the stability of superhydrophobic performance is a major problem to be solved [11].

Afrin [12] developed a simple synthesis method with two steps, using 3-glycidoxypropyl-trimethoxysilane (GPTMS) to coat the substrate and form epoxy nanoparticles to increase the surface roughness and 1H,1H,2H,2H perflurocotyl-trichlorosilane (TCFs) to reduce the surface energy of the substrate, thus providing a coating with excellent mechanical and chemical stability. However, the cumulative toxicity and pollution of fluorine-containing compounds limit their application. Cheng [13] used the ultrasonic assisted in-situ growth method to deposit composite nanoparticles formed by cashew phenol and ethyl orthosilicate on a cotton fabric surface to prepare superhydrophobic cotton fabric with excellent oil–water separation efficiency, but its raw material cashew phenol is very expensive. Rius-Ayra et al. [14] prepared ferromagnetic CuFeCo powder combined with high-energy ball milling and liquid-phase deposition. The powder was functionalized with dodecanoic acid, which not only can remove microplastics but also has superhydrophobic properties. However, its preparation time was long, and the reaction condition was harsh, because it must be carried out in an argon atmosphere in order to prevent its oxidation.

Click chemistry has been widely used because of its high chemical selectivity, high yield, fast reaction speed, and easy formation of strong chemical bonds [15]. Among them, mercaptan ene click chemistry (enol click chemistry) is the reaction between mercaptan and an unsaturated carbon–carbon double bond. The reaction conditions are mild, and it can be reacted under atmospheric pressure [16].

In this work, the carbon–carbon double bond carried by the plant polyphenol caffeic acid itself was used to introduce sulfhydryl groups on the surface of PET through an enol click chemical reaction under UV radiation. Fe^2+^ was used to form a chelate with CA to increase the roughness of the fabric surface, and superhydrophobic polyester fabric CA-Fe-SH-PET was prepared. The absorbed CA on the PET surface formed PCA under the UV radiation and adhered to the surface of PET, which provided robust stability for CA-Fe-SH-PET.

The formation process of chelate is shown in Figure 1a. The caffeic acid deposited on the surface of the fabric chelates with divalent metal iron ions to form a metal framework and oxidizes to ferric ions. The long carbon chain n-octadecyl mercaptan was linked to the chelates by a click chemical reaction through a photoinitiator, which is shown in Figure 1b. Under the irradiation of ultraviolet light, the photoinitiator benzoin dimethyl ether splits to generate free radicals, which obtain hydrogen atoms on the mercapto group and make it a mercapto radical. The mercapto radical attacks the carbon–carbon double bond in the caffeic acid structure to generate alkyl radicals. Next, the alkyl radical attacks the hydrogen atoms on the mercapto group to generate new mercapto radicals and new free radical chains.

An infrared spectrometer (IR), an energy dispersive spectrometer (EDS), and X-ray photoelectron spectroscopy (XPS) were used to characterize the chemical structure and elemental composition of the PET fabric. The surface morphology of the fabric was observed by scanning electron microscopy (SEM). The fabric was put in simulated harsh environments, and its hydrophobic stability was tested to prove its potential application value.

## 2. Materials and Methods

### 2.1. Materials

Polyester fabric (plain fabric, 89 gm^−2^, warp density: 40 threads/cm; weft density: 27 threads/cm), bought from a market, was washed with soap solution before use. Caffeic acid, n-octadecyl mercaptan, ferrous sulfate heptahydrate, and oil red O, were all purchased from Shanghai Aladdin Biochemical Technology Co., Ltd., Shanghai, China Cyclohexane (CYH), carbon tetrachloride (CCl_4_), methylene chloride (MC), tetrhydrofuran (THF), n-heptance (N-H), methanol (MT), and Ever Acid Blue N-RL dyes were all bought from Jiangsu Qiangsheng Functional Chemistry Co., Ltd., Suzhou, China. Diesel oil (Cyrus) was purchased from Zhengzhou Mingxin Lubricating Oil Co., Ltd., Zhengzhou, China. All other chemicals were analytical reagent grade and were not further purified when used.

### 2.2. Preparation of Superhydrophobic Polyester

As shown in Figure 2, first, 2 g PET fabric was put into a flask with 40 mL of deionized water solution containing 2 g/L of caffeic acid, and the temperature was raised to 80 °C for 15 min to obtain CA-PET fabric. Then, 6 g/L ferrous sulfate heptahydrate was added and fully reacted for 45 min to form chelates with CA absorbed onto the PET fabric surface. Afterwards, the fabric was taken out and placed in an oven at 60 °C for 30 min to form CA-Fe-PET fabric. Then, 3.2 g/L n-octadecyl mercaptan and 0.02 g/L sabbath dimethyl ether (DMPA) were dissolved in ethanol, and the liquid in the flask was poured into the ethanol solution. The CA-Fe-PET fabric was immersed into the flask and reacted for 1 h under UV light (220 W, 50 Hz). Finally, the super hydrophobic CA-Fe-SH-PET fabric was prepared by washing with water and drying in an oven at 60 °C for 6 h.

### 2.3. Characterization and Analysis

The surface morphology of the fabrics before and after modification were observed by Scanning Electron Microscope (SEM, Hitachi 8100) at 3.0 kv. The chemical structure of the fabric surfaces was analyzed using attenuated total reflectance-Fourier transform infrared spectroscopy (ATR-FTIR, German Brooke vertex 70 + hyperion 2000 model) in the range of 4000–500 cm^−1^ at a resolution of 4 cm^−1^.

The element composition of the sample surface was observed using the energy spectrum accessories of the scanning electron microscope X-ray spectrometer (EDS) and X-ray photoelectron spectroscopy (XPS, American Thermo Scientific k-alpha model).

### 2.4. Stability Test

The stability of the superhydrophobic CA-Fe-SH-PET fabric includes strength stability, friction stability, washing stability, UV stability, and chemical stability (seawater resistance stability, pH stability, and organic reagent stability), which were tested according to our previous work [17].

### 2.5. Oil–Water Separation Performance Test

Heavy oil chlorobenzene marked with oil red O and water marked with Everacid Blue N-RL were separated through a self-made gravity-driven oil–water separation device with the CA-Fe-SH-PET fabric fixed between the glass tube and the flask. The ratio of oil to water was 1:1. A polyurethane nano sponge coated with CA-Fe-SH-PET fabric was used as an adsorption bag to separate light oil cyclohexane. The weight of the adsorption bag was 1.76 g.

The oil–water separation efficiency is calculated according to Equation (1):(1)$$ŋ\left(\%\right)=\frac{V_{1}}{V_{0}}\times 100\%$$
where *V*_1_ and *V*_0_ are the volume of water before and after oil–water separation, and the unit is L.

The separation flux is calculated according to Equation (2):(2)$$\mathrm{Flux}\;=\frac{V}{\left(S\times t\right)}$$
where *V* is the volume of liquid passing through (*L*); S is the effective passing area (m^2^); *t* is the time through the liquid (h).

### 2.6. Photothermal Conversion Performance Test

In order to study the photothermal properties of the CA-Fe-SH-PET fabric, a 4 × 10 cm piece of the CA-Fe-SH-PET fabric was tied on the glass beaker with a rubber band, and a stable simulated UV lamp (250 W UV high-pressure mercury lamp) was used. After the light source was stable, the CA-Fe-SH-PET fabric was irradiated for 0, 0.5, 1,2, 4, 8, 12, and 16 min, respectively. A thermal infrared imager (E6, Philips, Seattle, WA, USA) was used to observe the temperature change of the fabric surface, and 5 drops of diesel oil were added to the surface of the CA-Fe-SH-PET fabric to observe the diffusion time.

### 2.7. UV Resistance Test

The sample PET, CA-PET, CA-Fe-PET, and CA-Fe-SH-PET fabrics were made into 10 × 10 cm-sized pieces, and UV protection factor (UPF) and UV transmittance of the fabrics were measured using a UV-2000 fabric UV protection tester (UltraScan PRO, Hunter Lab, Reston, VA, USA) according to the standard GB/T 18830-2009. The average of four tests on a single layer fabric was taken as the measurement result.

## 3. Results and Discussion

### 3.1. Surface Morphology and Chemical Structure Characterization

The surface morphology of the PET fabric, the CA-PET fabric, the CA-Fe-PET polyester fabric, and the CA-Fe-SH-PET fabric are shown in Figure 3. It can be seen from Figure 3a that the surface of untreated polyester fabric is relatively smooth, and the contact angle is 72°; the SEM image of CA-PET in Figure 3b shows that CA absorbed on the fabric had a sparse blocklike structure. The CA-Fe-PET treated with caffeic acid and a divalent iron ion, as shown in Figure 3c, formed uniform chelate particles like grapes on the surface of the fabric, forming a rough surface. The surface morphology of the polyester fabric (SH-PET) directly treated with mercaptan is shown in Figure 3d. The surface of the fabric was sparse needle-like, and the static contact angle of the fabric surface reached 126°, which is hydrophobic but not superhydrophobic. Through mercaptan ene click chemical reaction, n-octadecyl mercaptan was grafted onto the surface of CA-Fe-PET. The surface morphology of CA-Fe-SH-PET is shown in Figure 3e. CA further polymerized under UV radiation and formed dense grapelike granular particles with chelates and low surface energy substance mercaptan, and the surface of the CA-Fe-SH-PET became black. The static contact angle of the CA-Fe-SH-PET fabric was as high as 160.3°, and the rolling angle was 3.9°, which has a superhydrophobic property. It is obvious from Figure 4a,b that there were only C and O elements on the surface of the polyester fabric before modification. After finishing, Fe and S appeared on the fabric surface, and the elements’ distribution was relatively uniform. SEM and EDS results showed that n-octadecyl mercaptan was grafted onto the surface of CA-Fe-PET with a rough structure by click chemical reaction to prepare the CA-Fe-SH-PET superhydrophobic fabric.

In order to further determine the chemical structure and surface composition of CA-Fe-SH-PET, FTIR and XPS were tested. As shown in Figure 5a, the peaks at 1711 cm^−1^, 1242 cm^−1^, and 1092 cm^−1^ of the PET fabric belonged to the stretching vibration of C=O, C–O (ester) and −CH_2_, respectively. The wide absorption band at 3300 cm^−1^ was assigned to the stretching vibration of the hydroxyl of the PET fabric and CA on the fabric surface, which is obvious for CA-PET absorbing a large amount of CA. The characteristic peaks at 1017 cm^−1^ and 872 cm^−1^ can be attributed to the bending vibration in the plane of the benzene ring and the antisymmetric stretching vibration of C–C. Two obvious C=C characteristic peaks appeared at 1640 cm^−1^ and 1619 cm^−1^ in CA-PET and CA-Fe-PET. After finishing with octadecyl mercaptan containing a long carbon chain, SH-PET and CA-Fe-SH-PET both appeared as two obvious −CH_3_ and −CH_2_ characteristic peaks at 2915 cm^−1^ and 2847 cm^−1^, whereas the 1640 cm^−1^ and 1619 cm^−1^ peaks disappeared for CA-Fe-SH-PET, indicating that mercaptan was grafted onto the PET through PCA adhering to the fabric by click chemical reaction. Since the SH peak is usually weak, no obvious characteristic peak was detected [18,19,20,21,22,23].

Figure 5b–f shows the C1s high-resolution spectra of the PET and different modified polyester fabrics. All samples showed the characteristic peaks of C–C/C–H (284.7 ev), C–O (286 ev), and C=O (288.6 ev). C=C (285 ev) appeared in CA-PET and CA-Fe-PET, which was derived from the unsaturated double bond in CA’s structure, indicating that caffeic acid was finished on the PET fabric. In addition, C–S (286 ev) peaks appeared on SH-PET and CA-Fe-SH-PET, whereas the double bonds on CA-Fe-SH-PET disappeared. Combined with Table 1, it can be concluded that the carbon atom content on the surface of the PET fabrics treated with n-octadecyl mercaptan increased significantly. Particularly, the carbon atom content on the surface of the CA-Fe-SH-PET fabric increased from 74.8% to 85.03%, and new elements (sulfur atom content 3.68%) appeared, which further indicates that the enol click chemical reaction occurred on the surface of the fabrics. The long carbon chain material was successfully grafted onto the polyester fabric, reducing the surface energy of the fabric [24,25,26,27,28].

### 3.2. Stability

Table 2 lists the breaking strength and elongation at a break in the polyester fabric. The cutting specifications were 5 × 30 cm. The PET fabric and the CA-FE-SH-PET fabric had three warp and weft pieces, respectively, which were stretched under the same conditions. The average value was taken for three tests. It can be seen that compared with the PET, the breaking strength of the CA-Fe-SH-PET fabric in the weft direction decreased slightly, but still remained above 350 N. Caffeic acid and n-octadecyl mercaptan were polymerized and closely covered on the surface of the polyester fibers, and thus the breaking strength in the warp direction increased by 200 N. The breaking elongation was almost unchanged in the weft direction, and it slightly decreased in the warp direction.

After the stability test, three different parts of the fabric were taken to test the WCA and the SA, and the average value was taken. As shown in Figure 6a, after continuous washing for 225 min, the WCA of CA-Fe-SH-PET fabric remained above 150°, and the SA was kept below 7°, and it still had super hydrophobicity. When working in sea water, various friction effects cannot be avoided. The modified fabric was rubbed for 200, 400, 600, 800, and 1000 cycles under 9 kPa pressure, and the contact angle and rolling angle of the fabric were measured. The results are shown in Figure 6b. With the increase in abrasion cycles, it can be seen that the fuzz on the fabric surface became longer. After friction, the WCA of the CA-Fe-SH-PET fabric remained above 159°, and the SA was below 10°, which still indicated a superhydrophobic function. Fabrics used in natural seawater will be affected by many harsh environments. Therefore, a fabric corrosion test in simulated seawater was carried out. Figure 6c shows that with the increase in immersion time in the seawater, the WCA of the CA-Fe-SH-PET fabric was almost unchanged, and the SA remained below 5° and still maintained superhydrophobic properties, indicating that the CA-Fe-SH-PET fabric has excellent seawater corrosion resistance. The UV intensity of seawater is high, which has a great impact on the fabric. Figure 6d shows that under the simulated UV lighting conditions, even after 24 h of illumination, the WCA of the CA-Fe-SH-PET fabric remained above 160°, and the SA was at about 3°, demonstrating that the modified PET fabric has good UV resistance. Marine oil pollution has different pH levels; thus, the CA-Fe-SH-PET fabric was immersed in different pH solutions to evaluate the acid and alkali resistance. From Figure 6e, it can be seen that the contact angle of the fabric was the best under neutral conditions. Under acidic or alkaline conditions, the WCA of the fabric slightly decreased and remained above 150°. The SA increased significantly under strong alkaline conditions, but was still below 9°, indicating that the CA-Fe-SH-PET fabric has good acid and alkali resistance. The stability test of an organic reagent can simulate the superhydrophobic stability of the modified fabric when it adsorbs various marine oils. Herein, five typical organic reagents were selected, namely, CCl_4_, dichloromethane (DCM), methanol (MT), tetrahydrofuran (THF), and n-heptane (n-H). After soaking for 72 h, WCA of the CA-Fe-SH-PET fabric (Figure 6f) was maintained above 150° and SA was below 6°, which is stable in organic reagents.

### 3.3. Anti-Fouling and Self-Cleaning Performance of CA-Fe-SH-PET

Figure 7a–d shows the anti-fouling properties of CA-Fe-SH-PET. It can clearly be seen that water droplets, coke, milk, and tea can permeated through the original PET fabric. This is because the large pore size on the surface of the PET fabric makes the PET fabric quickly absorb the coke, milk, and tea, and the fabric surface is polluted. However, these droplets existed stably on the surface of CA-Fe-SH-PET, indicating that the adhesion of the CA-Fe-SH-PET fabric is very low, and the droplets adhered to the fabric surface. It can be seen from Figure 7e that vegetable oil can be instantly absorbed and penetrates through the surface of the CA-Fe-SH-PET, showing its lipophilicity. When the CA-Fe-SH-PET was completely immersed in water, it can be seen in Figure 7f that a “mirror” phenomenon formed on the fabric surface, and emission occurred on the superhydrophobic CA-Fe-SH-PET surface. An isolating layer with a large number of bubbles formed on the CA-Fe-SH-PET surface, which remained completely dry after being taken out of the water [29]. Figure 7g shows the effect of water spray on the CA-Fe-SH-PET fabric. It can be seen that the water was scattered and landed around the fabric without sticking to the fabric surface.

Self-cleaning is the most prominent feature of superhydrophobic surfaces. As shown in Figure 8a–c, when the pollutants (Bengal rose dye) on the original PET fabric surface were washed by water drops, the PET surface directly adsorbed the water-soluble pollutants and was polluted. The water droplets of pollutants had an ultra-low adsorption performance on the superhydrophobic surface. After washing with clean water, the water droplets could directly remove the pollutants and make them quickly leave the fabric surface, so that the fabric surface can be clean without leaving traces.

The adsorption of the modified fabric on oil with different carbon chains was further intuitively tested, and the results are shown in Figure 9a–c. Four to five drops of labeled heavy oil chlorobenzene were dropped into clean water with a rubber tip dropper, and it can be clearly observed that the heavy oil quickly adsorbed on the CA-FE-SH-PET fabric surface until it was completely “stuck” to the fabric surface. In Figure 9d–f, cyclohexane was selected as an example of light oil, and the modified fabric was used to remove the adhesion to achieve the effect of cleaning light oil.

A gravity experiment is shown in Figure 10a–c. In the process, an oil–water mixture with a volume ratio of 1:1 was poured into a self-weight device. When the complete separation was achieved, the separation flux and separation efficiency were calculated. The experiment was repeated 30 times. It can be seen from Figure 10g that after 30 cycles, the separation flux was still above 1200 Lm^−2^ h^−1^, and the separation efficiency hardly decreased. As shown in Figure 10d–f, the modified fabric was wrapped with a nano sponge to make an adsorption package. The adsorption package was used to adsorb the oil–water mixture of light oil. After 30 cycles, it can be seen from Figure 10h that the separation efficiency was still above 95%, and the stability effect was excellent.

### 3.4. Photothermal Performance

It can be seen from Figure 11a that the CA-Fe-SH-PET fabric has unique photothermal conversion characteristics. Under the same conditions, it took 147 s for heavy diesel oil to change from its initial state in Figure 11d to the full diffusion state in Figure 11e when the PET fabric was not exposed to ultraviolet light. As shown in Figure 12, compared with the PET fabric, the temperature of the CA-Fe-SH-PET fabric rose rapidly under short-term ultraviolet radiation. Figure 12g shows that after irradiation for 1 min, the surface temperature of the CA-Fe-SH-PET fabric rose to 36 ℃, and it took 61 s for heavy diesel oil to change from the initial condensed viscous state in Figure 11b to the full diffusion state in Figure 11c. When the irradiation time increased to 2 min, the CA-Fe-SH-PET fabric surface increased to 40.9 ℃ (Figure 12g), and the diffusion time of the diesel oil on the fabric surface decreased to 42 s. When the irradiation time increased to 16 min, the CA-Fe-SH-PET fabric surface reached 70.3 ℃ (Figure 12j), which was 30 ℃ higher than the original fabric. In addition, the diffusion time of the diesel on the CA-Fe-SH-PET fabric surface can be reduced to 22 s, showing excellent photothermal performance. First, this is due to the polymerization of caffeic acid and the formation of melanin chelated with Fe^2+^, which can convert absorbed ultraviolet radiation energy into thermal energy and provide a local warming effect [29,30]. As far as we know, the viscosity of diesel engine oil is inversely related to the temperature. As the temperature rises, the viscosity of the diesel engine oil decreases, and its diffusion rate is accelerated. In addition, polycaffeic acid molecule contains a large number of benzene rings with an obvious π–π conjugation effect, which can be responsible for its excellent light absorption characteristics in the visible–near UV region [31]. It can convert the absorbed ultraviolet light energy into thermal energy.

### 3.5. UV Resistance

It can be seen from Figure 13 that the UPF of the PET, CA-PET, and CA-Fe-PET fabrics were all below 30, and the UVA and UVB values of the PET and CA-PET fabrics were above 5, which indicated no UV resistance. However, the UPF of the CA-Fe-SH-PET fabric was up to 95.98, and both the UVA and the UVB were below the value of 2, far exceeding the standard of “UV protection products”(UPF value is greater than 40, UVA value is less than 5%). This was also derived from the polymerization of caffeic acid and the chelation reaction between caffeic acid and divalent iron ions on the surface of the polyester fabric, and the dark-brown chelation product produced was tightly coated on the polyester fabric to absorb ultraviolet light. In addition, due to the click chemical reaction under UV irradiation, the introduced thiol reduced the interstices between fibers and prevented the penetration of UV rays.

## 4. Summary

Based on caffeic acid and *n*-octadecyl mercaptan, a superhydrophobic/lipophilic CA-Fe-SH-PET fabric was fabricated by enol click chemical reaction. The prepared fabric had excellent superhydrophobic properties, as well as self-cleaning and antifouling properties, and it could maintain physical and chemical stability in different harsh environments. After 30 separation cycles, the CA-Fe-SH-PET fabric still maintained high separation efficiency, showing a potential application prospect in oil–water separation. Furthermore, the CA-Fe-SH-PET fabric had a good photothermal conversion performance. It converted absorbed light into thermal energy in a short time and quickly absorbed and diffused viscous diesel oil. The CA-Fe-SH-PET fabric also had excellent UV resistance, with a UPF of 95.98, which is more conducive to its application in oil–water separation.

## Figures and Tables

**Figure 1 polymers-14-05536-f001:**
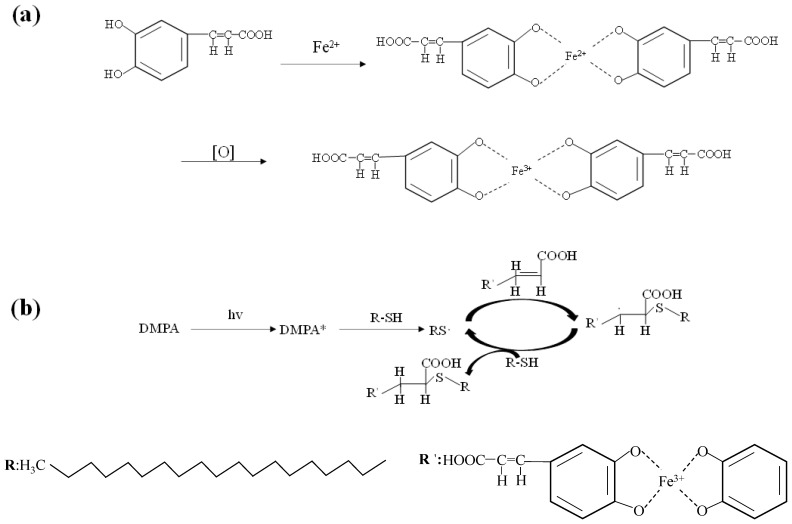
(**a**) Caffeic acid forms chelate with iron ions; (**b**) click chemical reaction of chelate with n-octadecyl mercaptan. (* is the state in which the photoinitiator splits into free radicals).

**Figure 2 polymers-14-05536-f002:**
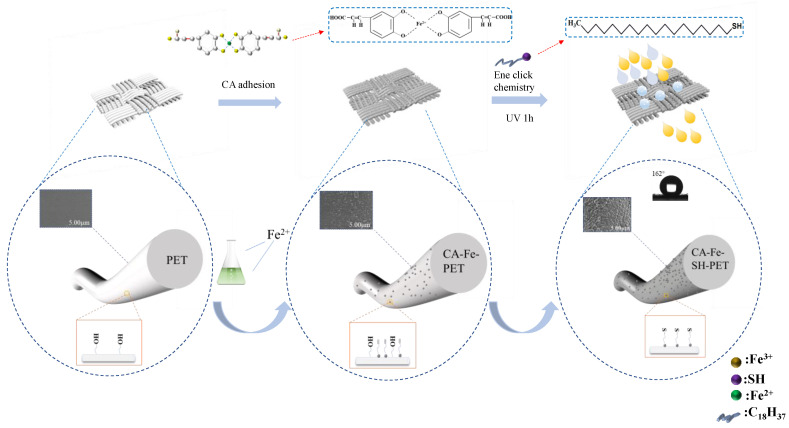
Preparation process for superhydrophobic polyester fabric.

**Figure 3 polymers-14-05536-f003:**
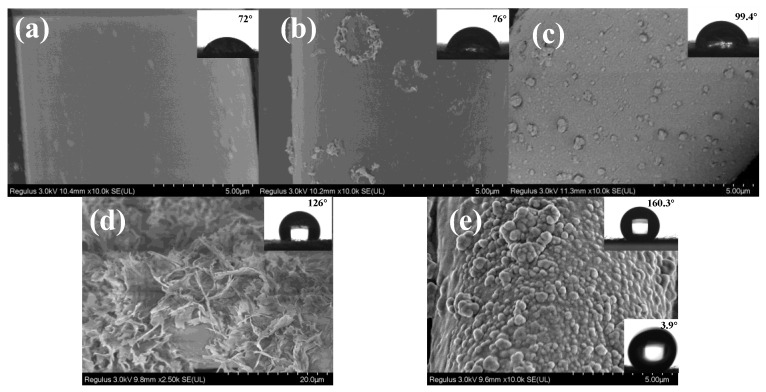
SEM photos of polyester fabric. (**a**) PET; (**b**) CA-PET; (**c**) CA-Fe-PET; (**d**) SH-PET; (**e**) CA-Fe-SH-PET.

**Figure 4 polymers-14-05536-f004:**
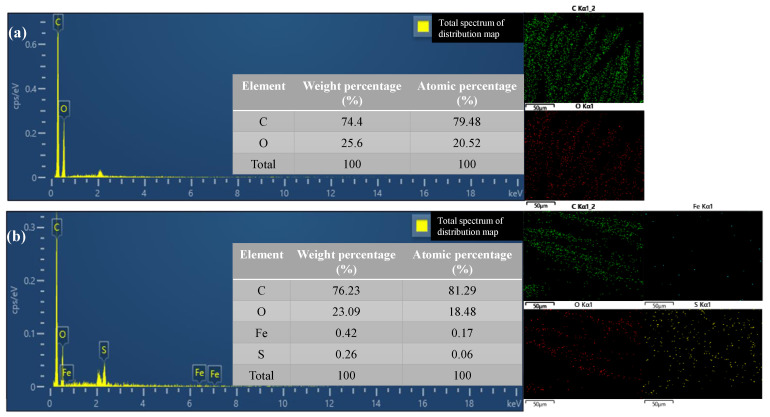
EDS spectra of (**a**) PET fabric and (**b**) CA-Fe-SH-PET fabric.

**Figure 5 polymers-14-05536-f005:**
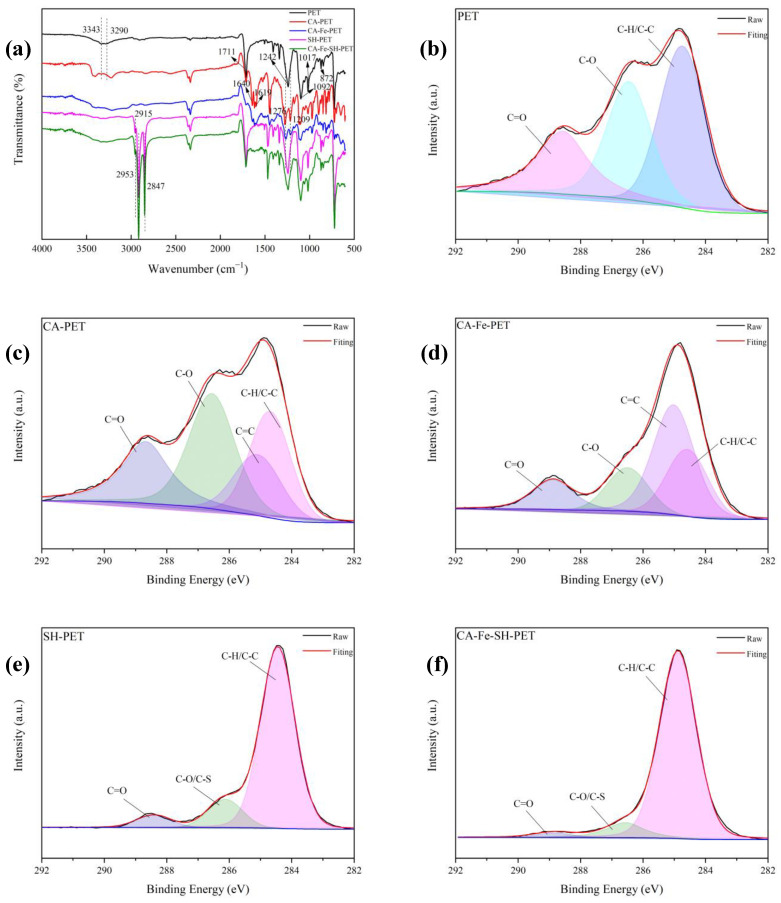
(**a**) ATR spectra of PET, CA-PET, CA-Fe-PET, SH-PET, CA-Fe-SH-PET. (**b**) High-resolution XPS spectra C1_S_ of PET, (**c**) CA-PET, (**d**) CA-Fe-PET, (**e**) SH-PET, and (**f**) CA-Fe-SH-PET.

**Figure 6 polymers-14-05536-f006:**
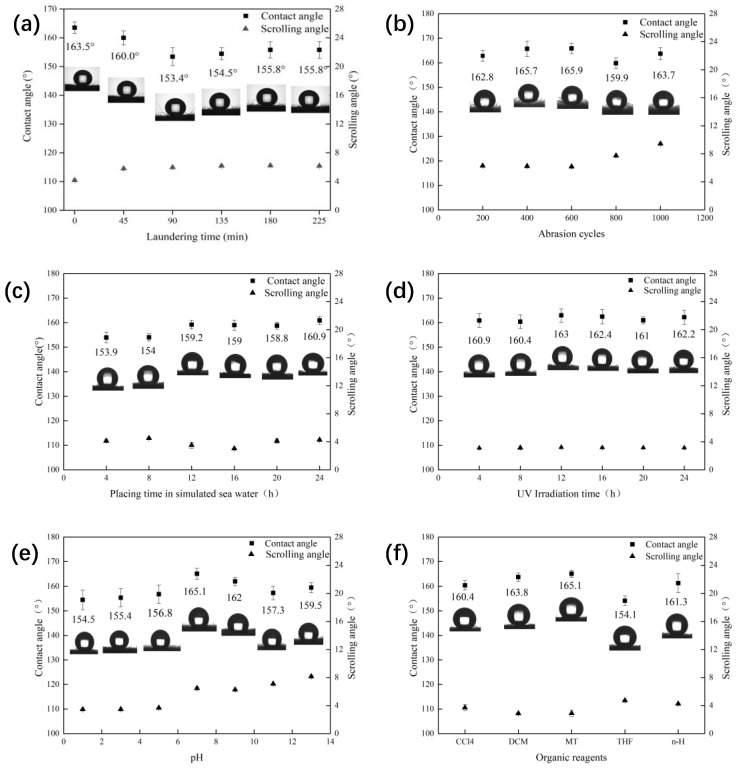
Superhydrophobic stability of CA-Fe-SH-PET fabric. (**a**) Water washing resistance; (**b**) friction resistance; (**c**) seawater resistance; (**d**) sun aging resistance; (**e**) pH resistance; (**f**) organic reagent resistance.

**Figure 7 polymers-14-05536-f007:**
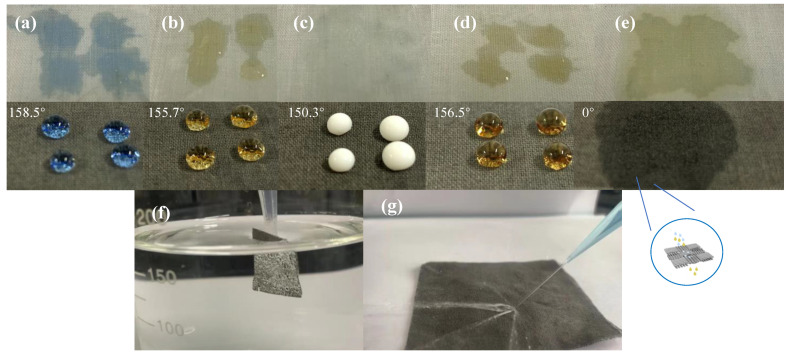
Droplets of Everacid Blue N-RL dyed water (**a**) cola (**b**), milk (**c**), tea (**d**), and vegetable oil (**e**) on the surface of original PET (upper) and CA-Fe-SH-PET (under); state of CA-Fe-SH-PET fabric in water (**f**); state of water spray on CA-Fe-SH-PET fabric (**g**).

**Figure 8 polymers-14-05536-f008:**
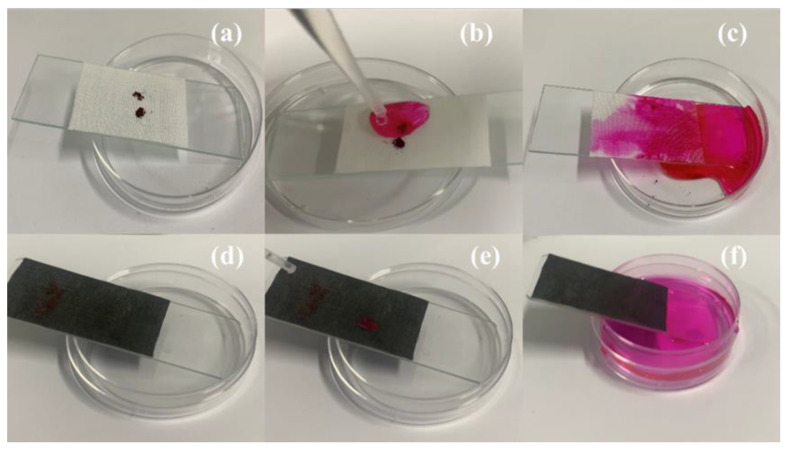
Self-cleaning performance of (**a**–**c**) original PET, (**d**–**f**) CA-Fe-SH-PET.

**Figure 9 polymers-14-05536-f009:**
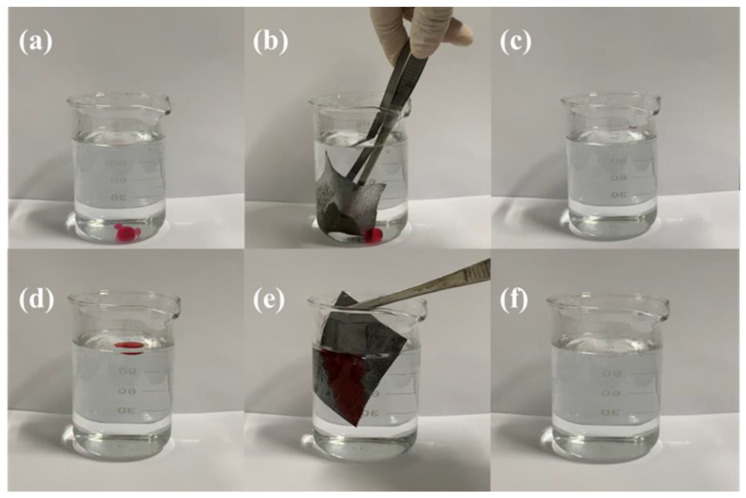
Selective absorption of CA-Fe-SH-PET fabric for heavy oil chlorobenzene (**a**–**c**) and light oil cyclohexane (**d**–**f**).

**Figure 10 polymers-14-05536-f010:**
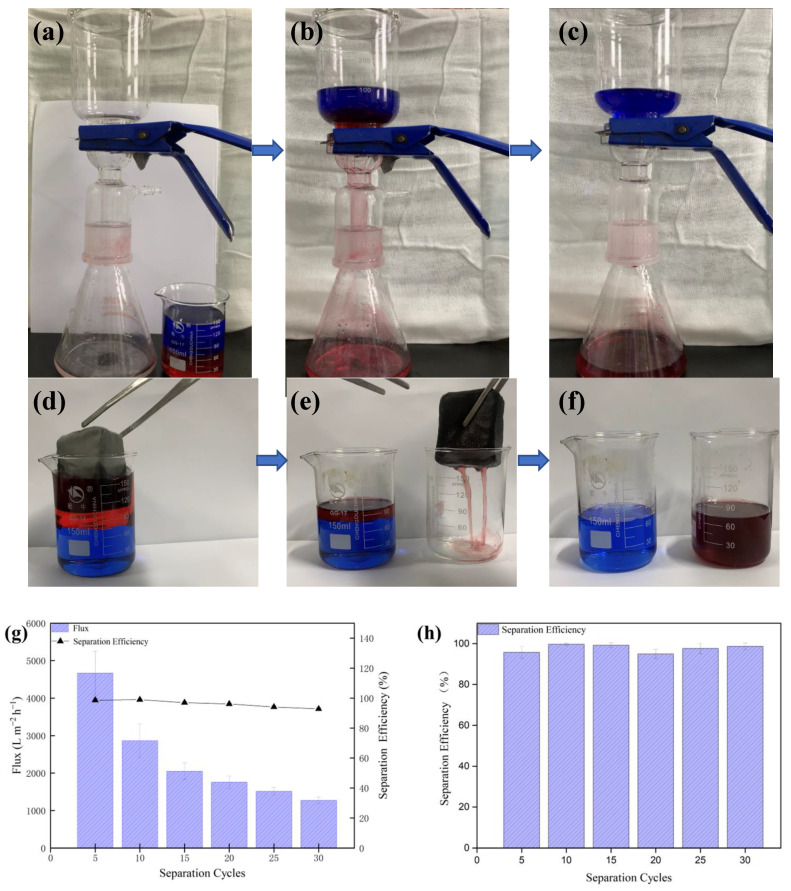
Oil–water separation performance of CA-Fe-SH-PET fabric. (**a**–**c**) Heavy oil separation by gravity-driven method; (**d**–**f**) light oil separation by adsorption bag; (**g**) heavy oil separation flux and separation efficiency; (**h**) light oil separation efficiency and recyclability.

**Figure 11 polymers-14-05536-f011:**
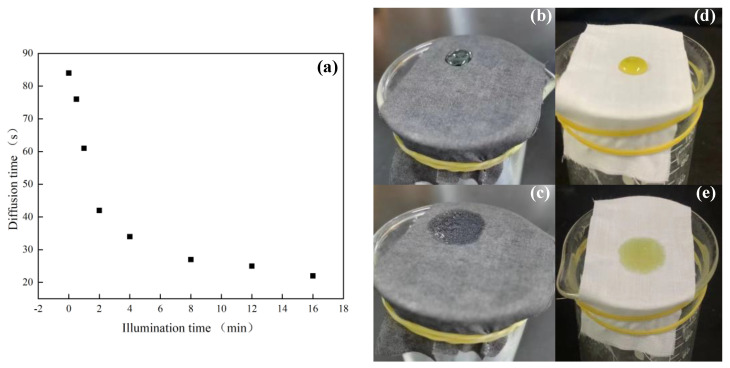
How quickly the CA-Fe-SH-PET fabric adsorbs diesel oil (**a**); state of diesel oil contacting the CA-Fe-SH-PET fabric (**b**); state of diesel oil completely adsorbed by the CA-Fe-SH-PET fabric (**c**); state of diesel oil contacting the PET fabric (**d**); state of diesel oil completely adsorbed by the PET fabric (**e**).

**Figure 12 polymers-14-05536-f012:**
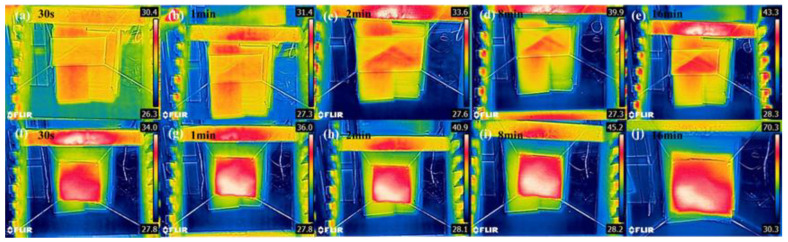
Thermal infrared image of PET after short-term ultraviolet radiation of PET fabrics (**a**–**e**) and CA-Fe-SH-PET fabric (**f**–**j**).

**Figure 13 polymers-14-05536-f013:**
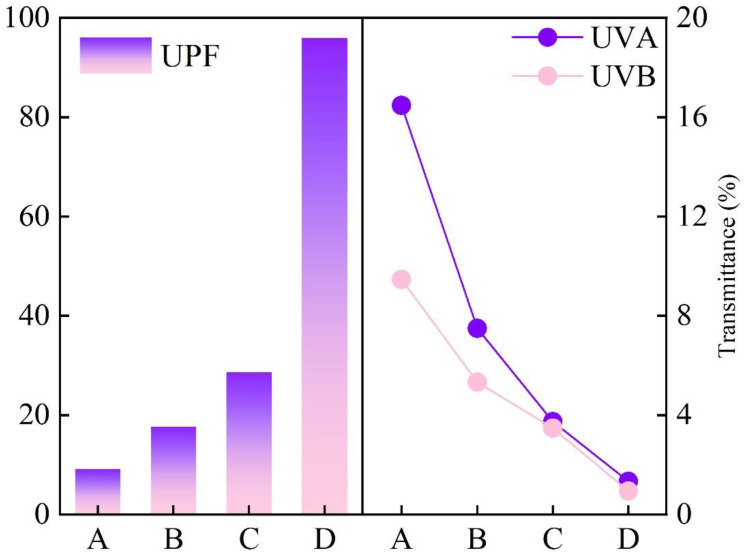
UV resistance of PET (A), CA-PET (B), CA-Fe-PET (C), and CA-Fe-SH-PET (D) fabrics.

**Table 1 polymers-14-05536-t001:** Atomic element percentages of PET, CA-PET, CA-Fe-PET, SH-PET, and CA-Fe-SH-PET.

Atomic Content (%)	PET	CA -PET	CA-Fe-PET	SH-PET	CA-Fe-SH-PET
C1s	74.8	71.24	71.23	79.83	85.04
Fe2p			1.29		0.56
S2p				3.03	3.69

**Table 2 polymers-14-05536-t002:** Breaking strength and elongation at break of PET fabrics.

Property	Original PET Fabric	CA-Fe-SH-PET Fabric
Breaking strength (N)	382.48 ± 25.50 (weft) 392.54 ± 20.41 (warp)	370.74 ± 16.24 (weft) 581.55 ± 11.86 (warp)
Elongation at break (%)	16.27 ± 0.81 (weft) 17.08 ± 0.32 (warp)	16.52 ± 0.77 (weft) 14.43 ± 0.22 (warp)

## Data Availability

The raw/processed data required to reproduce these findings cannot be shared at this time, as the data also form part of an ongoing study.

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
