# Peer review of "Fabrication of Superhydrophobic and Light-Absorbing Polyester Fabric Based on Caffeic Acid"

_polymers, 2022, doi:10.3390/polym14245536_

Round 1

Reviewer 1 Report

Article “Fabrication of superhydrophobic and light absorbing polyester fabric based on caffeic acid” is quite interesting and corresponds to the current trends. However, the article requires correction and clarification of the obtained results. Below is attached a list of major and minor remarks that need to be fulfilled and revised in order for the work to be accepted for publication.

Figure 1 and its description should be moved to objects and methods of study. The description for Figure 1 should be more extended.

Lines 76 to 80 check the layout.

The SEM in Figure 2 is not necessary.

For CA-Fe-SH-PET, that in Figure 2 reflects the -SH end groups on the surface of the fibre, which is incorrect with respect to your chemical reaction scheme.

Section 3.1 can be reformatted for readability in the system:

SEM description

   Figure SEM

FTIR description

   Figure FTIR, etc.

In the FTIR description section, the highlighted peaks need to be supported by references to the literature. Similar to these articles:

10.1007/s10973-021-10895-z

10.3390/polym14214594

Section 3.5 can be improved with data on the effect of UV radiation on the diffusion times of other fluids through CA-Fe-SH-PET fabric.

Line 319-321. You can't say about maintain mechanical properties of polyester fabric without tests after the fabric staying in a harsh environment.

Author Response

First of all, thank you for your valuable comments on this article. In addition, I have implemented the suggestions and responded to the comments of the reviewers.

All the changes and new parts are highlighted in red color in the text.

Thanks for your reconsideration.

Response to Reviewer 1 Comments

  1. Figure 1 and its description should be moved to objects and methods of study. The description for Figure 1 should be more extended.

Response: Thanks for your suggestion. The descripiton was extended and added in the text as follows:

The formation process of chelate is shown in Fig. 1 (a). The caffeic acid deposited on the surface of the fabric chelates with divalent metal iron ions to form a metal framework and oxidizes to ferric ions. The long carbon chain n-octadecyl mercaptan was linked to the chelates by click chemical reaction through photoinitiator, which is shown in Fig. 1 (b). Under the irradiation of ultraviolet light, the photoinitiator benzoin dimethyl ether splits to generate free radicals, which obtain hydrogen atoms on the mercapto group and make it a mercapto radical. The mercapto radical attacks the carbon carbon double bond in the caffeic acid structure to generate alkyl radicals. Next, the alkyl radical attacks the hydrogen atoms on the mercapto group to generate new mercapto radicals and new free radical chains.

  1. Lines 76 to 80 check the layout.

Response: Thank you. We have corrected in red in the original text.

  1. The SEM in Figure 2 is not necessary.

Response: The electron microscope pictures are put in order to express the actual changes before and after fabric surface finishing.

  1. For CA-Fe-SH-PET, that in Figure 2 reflects the -SH end groups on the surface of the fiber, which is incorrect with respect to your chemical reaction scheme.

Response: Thank you. In the original picture, we will remove the hydrogen atom on the sulfhydryl group after clicking the chemical reaction to form a C-S bond.

  1. Section 3.1 can be reformatted for readability in the system: SEM description、Figure SEM、FTIR description、Figure FTIR, etc.

Response: Thank you. We will improve the layout of the pictures in the original image and adjust the size of the pictures.

  1. In the FTIR description section, the highlighted peaks need to be supported by references to the literature. Similar to these articles:10.1007/s10973-021-10895-z、3390/polym14214594.

Response: Thank you! The Document supported literature has been added to the references.

  1. Section 3.5 can be improved with data on the effect of UV radiation on the diffusion times of other fluids through CA-Fe-SH-PET fabric.

Response: Thanks for your suggestion. However, diesel oil, engine oil and other viscous oil substances belong to pipe products, the laboratory cannot purchase such products for the time being and cannot further test them.

  1. Line 319-321. You can't say about maintain mechanical properties of polyester fabric without tests after the fabric staying in a harsh environment.

Response: Thanks for your advice. It is changed into: The prepared fabric has excellent superhydrophobic properties, self-cleaning and antifouling properties, and can maintain physical and chemical stability in different harsh environments.

Reviewer 2 Report

1-at page 5 line 157 in result and discussion part and at page 12 line 292 in the same part the sentence start with (And) you must replace it by in addition or suitable another phrase.

2-at page 7 line 210 the sentence not start with (because) you can put (,) vice fall stop.

3-at page 8 line 216 please clear WCA and SA.

4-where is the discussion on figures 9&10???

5-the part of photothermal performance need to be rewrite again because it is not clear.

Author Response

Dear reviewer 2,

First of all, thank you for your valuable comments on this article. In addition, I have implemented the suggestions and responded to the comments of the reviewers.

All the changes and new parts are highlighted in red color in the text.

Thanks for your reconsideration.

Response to Reviewer 2 Comments

  1. at page 5 line 157 in result and discussion part and at page 12 line 292 in the same part the sentence start with (And) you must replace it by in addition or suitable another phrase.

Response: Thanks for your suggestion. We have corrected the description. The first sentence was deleted. And for the second sentence, “In addition” was used to replace “And” the diffusion time of diesel on CA-Fe-SH-PET fabric surface can be reduced to 22s, showing excellent photothermal performance.

  1. at page 7 line 210 the sentence not start with (because) you can put (,) vice fall stop.

Response: Thanks for your advice. It was cleared as follows: Caffeic acid and n-octadecyl mercaptan were polymerized and closely covered on the surface of polyester fibers, thus the breaking strength in warp direction increases by 200N.

  1. at page 8 line 216 please clear WCA and SA.

Response: Thank you! As shown in Fig. 6 (a), after continuous washing for 225 minutes, the WCA of CA-Fe-SH-PET fabric is kept above 150 ° and SA is kept below 7 °, and it still has super hydrophobicity.

  1. where is the discussion on figures 9&10?

Response: Sorry for our negligence. The discussion was added in the manuscript as follows

The adsorption of the modified fabric on oil with different carbon chains was further intuitively tested, and the results are shown in Figure 9 (a) - (c). 4-5 drops of the labeled heavy oil chlorobenzene were dropped into the clean water with a rubber tip dropper, and it can be clearly observed that the heavy oil will quickly adsorb on the CA-FE-SH-PET fabric surface until it is completely "stuck" to the fabric surface. In Fig. 9 (d) - (f), cyclohexane is selected as the example of light oil, and the modified fabric is used to remove the adhesion to achieve the effect of cleaning light oil.

The gravity experiment is shown in Fig. 10 (a) - (c). In the process, the oil-water mixture with a volume ratio of 1:1 was poured into the self weight device. When the complete separation was completed, separation flux and separation efficiency was calculated. The experiment was repeated for 30 times. It can be seen from Fig. 10 (g) that after 30 cycles, the separation flux is still above 1200Lm-2h-1, and the separation efficiency hardly decreased. As shown in Figure 10 (d) - (f), the modified fabric is wrapped with nano sponge to make an adsorption package. The adsorption package was used to adsorb the oil-water mixture of light oil. After 30 cycles, it can be seen from Figure 10 (h) that the separation efficiency is still above 95%, and the stability effect is excellent.

  1. the part of photothermal performance need to be rewrite again because it is not clear.

Response: Thanks for your suggestion. This part was rewritten as follows:

As far as we know, the viscosity of diesel engine oil is inversely related to the temperature. As the temperature rises, the viscosity of diesel engine oil decreases and its diffusion rate is accelerated. In addition, polycaffeic acid molecule contains a large number of benzene rings with obvious π π conjugation effect, which can be responsible for its excellent light absorption characteristics in visible - near UV region.[31] It can convert the absorbed ultraviolet light energy into thermal energy.

Reviewer 3 Report

In abstract, water contact angle is short as WCA instead of WAC. 

In Figure 3d, magnification is different from other materials and should be updated to be consistent so that morphology is comparable. 

In Figure 6 and Section 3.2, number of experiments should be indicated for each of the conditions, and also need to indicate if the samples are statistically significant or based on single sample experiment. 

Figure 7f needs to have a higher resolution to demonstrate the bubbles clearly as indicated in the main text. Dimension references should also be added to all inset in Figure 7. 

Figures 9 and 10 are never referenced in the main text. Additional details should be added, or plotted should be removed to avoid confusion. 

Figure 11, treated fabric should be compared with untreated PET to demonstrate difference in diffusion behavior. Also, the photothermal behavior in Figure 11 should be explained in more details such as conditions of the illumination and diffusion behaviors. 

Figure 13, UVA and UVB are plotted but never referenced in the main text. Additional details should be added, or plotted should be removed to avoid confusion. 

Author Response

Dear Reviewer 3,

First of all, thank you for your valuable comments on this article. In addition, I have implemented the suggestions and responded to the comments of the reviewers.

All the changes and new parts are highlighted in red color in the text.

Thanks for your reconsideration.

Response to Reviewer 3 Comments

  1. In abstract, water contact angle is short as WCA instead of WAC.

Response: Sorry for our negligence. It was corrected in the text.

  1. In Figure 3d, magnification is different from other materials and should be updated to be consistent so that morphology is comparable.

Response: The picture in Figure 3 (d) has too large magnification, which will lead to unclear structure. The other pictures in Figure 3 have too small magnification, which will lead to unclear surface structure.

  1. In Figure 6 and Section 3.2, number of experiments should be indicated for each of the conditions, and also need to indicate if the samples are statistically significant or based on single sample experiment.

Response: Thanks for your suggestion. We have added the test conditions in the manuscript as follows:

The cutting specifications are 5 × 30 cm PET fabric and the CA-FE-SH-PET fabric have three warp and weft pieces respectively, which are stretched under the same conditions. The average value is taken for three tests.

After the stability test, take three different parts of the fabric to test WCA and SA, and take the average value. As shown in Fig. 6 (a), after continuous washing for 225 minutes, the WCA of CA-Fe-SH-PET fabric is kept above 150 ° and SA is kept below 7 °, and it still has super hydrophobicity.

  1. Figure 7f needs to have a higher resolution to demonstrate the bubbles clearly as indicated in the main text. Dimension references should also be added to all inset in Figure 7.

Response: Thanks for your suggestion. We have replaced Figure 7(f) with higher resolution. And other images in Figure 7 are just photos by taken general camera, so we did not put dimension references.

  1. Figures 9 and 10 are never referenced in the main text. Additional details should be added, or plotted should be removed to avoid confusion.

Response: Sorry for our negligence. The discussion was added in the manuscript as follows

The adsorption of the modified fabric on oil with different carbon chains was further intuitively tested, and the results are shown in Figure 9 (a) - (c). 4-5 drops of the labeled heavy oil chlorobenzene were dropped into the clean water with a rubber tip dropper, and it can be clearly observed that the heavy oil will quickly adsorb on the CA-FE-SH-PET fabric surface until it is completely "stuck" to the fabric surface. In Fig. 9 (d) - (f), cyclohexane is selected as the example of light oil, and the modified fabric is used to remove the adhesion to achieve the effect of cleaning light oil.

The gravity experiment is shown in Fig. 10 (a) - (c). In the process, the oil-water mixture with a volume ratio of 1:1 was poured into the self weight device. When the complete separation was completed, separation flux and separation efficiency was calculated. The experiment was repeated for 30 times. It can be seen from Fig. 10 (g) that after 30 cycles, the separation flux is still above 1200Lm-2h-1, and the separation efficiency hardly decreased. As shown in Figure 10 (d) - (f), the modified fabric is wrapped with nano sponge to make an adsorption package. The adsorption package was used to adsorb the oil-water mixture of light oil. After 30 cycles, it can be seen from Figure 10 (h) that the separation efficiency is still above 95%, and the stability effect is excellent.

  1. Figure 11, treated fabric should be compared with untreated PET to demonstrate difference in diffusion behavior. Also, the photothermal behavior in Figure 11 should be explained in more details such as conditions of the illumination and diffusion behaviors.

Response: As far as we know, the viscosity of diesel engine oil is inversely related to the temperature. As the temperature rises, the viscosity of diesel engine oil decreases and its diffusion rate is accelerated. In addition, polycaffeic acid molecule contains a large number of benzene rings with obvious π −π conjugation effect, which can be responsible for its excellent light absorption characteristics in visible - near UV region.[31] It can convert the absorbed ultraviolet light energy into thermal energy.

7、Figure 13, UVA and UVB are plotted but never referenced in the main text. Additional details should be added, or plotted should be removed to avoid confusion.

Response: The UVA and UVB values of PET and CA-PET fabrics are above 5, which have not UV resistance. However, the UPF of CA-Fe-SH-PET fabric is high to 95.98, and both UVA and UVB are below the value of 2, far exceeding the standard of "UV protection products". (UPF value is greater than 40, UVA value is less than 5%).

Round 2

Reviewer 1 Report

The revised article  “Fabrication of superhydrophobic and light absorbing polyester fabric based on caffeic acid” was well revised and may be accept for publication in present form.

Reviewer 3 Report

Thank you for making the edits and responses.